# Digital Transformation and Artificial Intelligence Applied to Business: Legal Regulations, Economic Impact and Perspective

**Ricardo Francisco Reier Forradellas *** 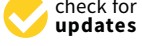 **and Luis Miguel Garay Gallastegui ***

Department of Economics, Catholic University of Ávila, 05005 Ávila, Spain
* Correspondence: ricardo.reier@ucavila.es (R.F.R.F.); lmiguel.garay@ucavila.es (L.M.G.G.)

**Abstract:** Digital transformation can be defined as the integration of new technologies into all areas of a company. This technological integration will ultimately imply a need to transform traditional business models. Similarly, artificial intelligence has been one of the most disruptive technologies of recent decades, with a high potential impact on business and people. Cognitive approaches that simulate both human behavior and thinking are leading to advanced analytical models that help companies to boost sales and customer engagement, improve their operational efficiency, improve their services and, in short, generate new relevant information from data. These decision-making models are based on descriptive, predictive and prescriptive analytics. This necessitates the existence of a legal framework that regulates all digital changes with uniformity between countries and helps a proper digital transformation process under a clear regulation. On the other hand, it is essential that this digital disruption is not slowed down by the regulatory framework. This work will demonstrate that AI and digital transformation will be an intrinsic part of many applications and will therefore be universally deployed. However, this implementation will have to be done under common regulations and in line with the new reality.

**Keywords:** digital transformation; organizational change; data analytics; artificial intelligence; regulatory normative



## 1. Introduction

For years now, we have been witnessing the transition from an analog society to a new digital society (new technologies, new business models, new ways of organizing and communicating, etc.) with the changes and transformations that this implies both in the way business models are conceived and in the new structuring of jobs.

Artificial intelligence (AI) is destined to be the next technological revolution, following in the footsteps of the Internet and mobility. Artificial intelligence (AI) can be understood as the simulation of human intelligence by machines and is defined as the ability of a machine to perform cognitive functions normally associated with the human brain, such as perceiving, reasoning, learning, evolving with experience, solving problems, interacting with the environment, and even exercising creativity.

In this scenario, it is necessary that the legislation that regulates all digital changes with uniformity between countries, such as the EU Directive 2000/31/EC—electronic commerce that establishes standard rules in the EU on various issues related to electronic commerce (EUR-Lex 2000), or the General Data Protection Regulation (GDPR) that regulates the protection of natural persons with regard to the processing of personal data and on the free movement of such data (Data Protection in the EU 2016), helps perform an adequate digital transformation process under clear regulations and, on the other hand, that this digital disruption is not slowed down by the regulatory framework. In the case of Spain, authors such as Castellote give as an example the regulations that agreed upon data protection and its regulation at the level of the General Data Protection Regulation—GDPR—as one of the legislative trends that must be taken into account to adapt the digital transformation

to the corresponding legal framework. The need to update the regulatory framework is even more necessary in the new digital environment, since obsolescence must also be applied to the regulations. Thus, the aforementioned GDPR in the case of Spain came to regulate a panorama that was marked by the Organic Law on Data Protection of 1995 (Castellote 2018).

In this sense, it must be taken into account that much of the previous legislation was made when social networks and many of the existing technological changes did not yet exist. This new legal regulation implies a regulation that must have a continuous process since companies must develop this obligatory compliance throughout their life cycle. Catellote himself points out that "any legislation that comes to the digital world must be a stable process. It cannot be a barrier to progress and must be a facilitator, but with guarantees that what is going to be done serves the interests of the industry, the global market and must be agreed locally with each EU country". Thus, and applying it to Spain, we can point out different regulations and different digitalization plans—also with the corresponding legal aspects—apply to different sectors in terms of digital transformation:

- Law 39 and 40/2015 for the digital transformation in the general administration of the State and its public bodies (Boletín Oficial del Estado 2015).
- Organic law for the protection of personal data and guarantee of digital rights (Boletín Oficial del Estado 2018).
- Law 7/2020 for the digital transformation of the financial system (Boletín Oficial del Estado 2020).
- Plan for the digitalization of public administrations (Government of Spain 2020).
- Digital Spain plan 2025 (Government of Spain 2021).

All these regulatory aspects must be taken into account in this process of digital disruption in which we are immersed, and especially with regard to artificial intelligence projects because they are based on data. Data are needed to train and validate the models. We also need to store and analyze the data generated by the interaction with the model since only then is it capable of determining whether the implementation of the solution meets the intended objectives. The constant evaluation of the models allows their improvement and evolution, and therefore, artificial intelligence projects are clearly influenced by current legislation on data protection.

This context is enormously suggestive, albeit too broad. Today, AI applied to the business world is more precisely oriented towards the discovery and analysis of information for the purpose of making predictions, recommendations and decision support; facilitating interactions with people; and automating certain responses. This orientation is very relevant in the field of digital transformation of companies and especially in terms of data-driven decision making. with an expected impact by 2030 that will represent a growth of 14% of global GDP (PWC 2017). AI will transform strategies and operating models of companies, with significant improvements in productivity models as an initial impact of its application. Estimates indicate that 45% of projected economic gains by 2030 may come from the commercial application of AI solutions applied to business. The Figure 1 shows the influence of digital transformation on the business environment.

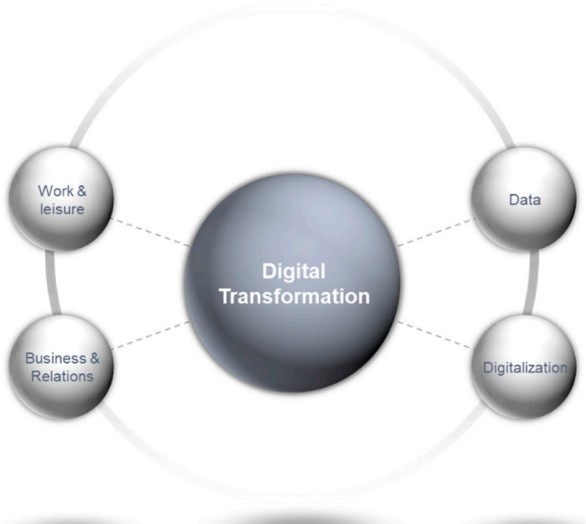

**Figure 1.** Digital transformation process in business. Source: Own elaboration.

The current possibilities of AI include capabilities such as the creation of simulation models or propensity to purchase, the personalization of the purchase process through recommendation systems based on machine learning technologies and also the interaction with virtual assistants to help with the purchase. All of these provide an excellent opportunity to, on the one hand, improve the customer experience with brands and, on the other hand, monetize this experience. We are thus entering a new stage of the Intelligent Experience Economy. Authors such as Bonnet and Westerman (2021) describe this experience in a global way for the company, digitally connecting all partners in the digital business ecosystem (Bonnet and Westerman 2021). This global dimension can be observed in the following Figure 2.

| BUSINESS MODEL | | |
|---|---|---|
| Digital enhancements | | |
| Information-based service extensions | | |
| Multisided platform businesses | | |
| **CUSTOMER EXPERIENCE** | **OPERATIONS** | **EMPLOYEE EXPERIENCE** |
| Experience design | Core process automation | Augmentation |
| Customer intelligence | Connected and dynamic operations | Future-readying |
| Emotional engagement | Data-driven decisión-making | Flexforcing |
| Digital Platform | | |
| Core | | |
| Externally facing | | |
| Data | | |

**Figure 2.** Digital Business Model. Source: Own elaboration based on Bonnet and Westerman (2021).

The development of Artificial Intelligence dates back to the second half of the 20th century when Alan Turing proposed what has been called the Turing Test: can a computer communicate well enough and intelligently enough so that a human cannot distinguish whether it is really a person or a machine? Turing proposed this test in his essay "Computing Machinery and Intelligence" (Turing 1950) while working at the University of

Manchester. This essay begins with the question "Can machines think?" and argues that it is possible to imagine computers that mimic human behavior (Kak 1995).

Turing's article has unquestionably generated more commentary and controversy than any other article in the field of artificial intelligence. Of course, the idea that machines can think and be intelligent is enormously attractive (Dreyfus 1992; Ginsberg 2012), but at the same time, the literal interpretation of the Turing Test has conditioned the construction of machines whose simple purpose is to overcome it by partially imitating the way of reasoning of humans.

Therefore, there are conflicting opinions on the validity of the Turing Test to demonstrate truly automated intelligence. Even from a practical point of view, machines could be designed that are specifically aimed at passing said Turing Test, but that would not mean that they are really intelligent. Consequently, the implications on the simulation of human thought and authentic intelligence generate debates that have sometimes even been considered detrimental to the advancement of artificial intelligence itself (Whitby 1996).

Despite the attractiveness of the proposal and subsequent work along these lines, artificial intelligence has gone through stages with ups and downs in its development and business application, sometimes having only the academic and research environment as a refuge. These times have been called AI winters and have usually been periods that have followed previous times of unfulfilled expectations.

However, in the 21st century, interest in AI has resurfaced again and, in this case, with the conviction not only of researchers but above all of investors and companies. Advances in machine learning, the availability of much more powerful computers and new capabilities in data processing have marked a turning point to the point of including AI not only as a self-learning technology but also a key part of digital transformation and the so-called fourth industrial revolution. The Figure 3 shows the different fields of application of artificial intelligence.

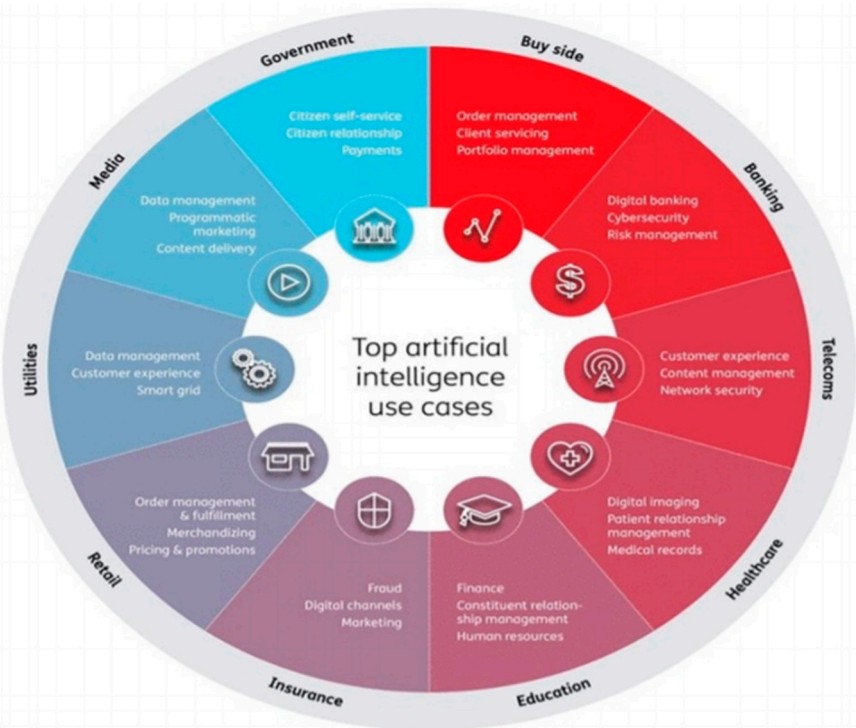

**Figure 3.** Top AI use cases. Source (Accenture 2017).

These AI use cases still have a long way to go, but the emergence of quantum computing will remove criticism about AI's constraints and current limitations. Quantum computing and artificial intelligence are transformational technologies, and the level of progress in artificial intelligence will accelerate exponentially with advances in quantum

computing. Although it is possible to develop functional artificial intelligence applications using classic computers, they are limited by the processing power of classic computers. Quantum computing can provide the quantum leap in computing terms that artificial intelligence needs to tackle more complex problems in many fields of business and truly intelligent reasoning (Dunjko and Briegel 2018). Logically, the definitions made from the academic field on the concept of digital transformation, first, and on artificial intelligence later, vary in specific aspects, but almost all of them coincide in their global nature. Several authors limit the impact of the application of different information technologies in organizational processes, considering it as just another evolutionary process (Heilig et al. 2017), while other academics directly define digital transformation as a total disruption in the business world (Skog et al. 2018). In fact, there are authors who go further and go so far as to define the process as the era of digital Darwinism (Solis 2018). It seems clear that any definition must be accompanied by "an evolutionary process that leverages digital capabilities and technologies to enable business models, operational processes and consumer experiences that generate value" (Morakanyane et al. 2017). Under this dimension, various authors have highlighted the need to link the digitalization process to all areas of the organization and under a new perspective defined as the fourth industrial revolution (Schwab 2016).

Thus, it is common in academia to link digitalization as a global maturity model that involves the implementation of new processes and models in all dimensions (processes, people, governance, etc.) (Rossmann 2018), or to link it directly to the capacity for interaction between user (people) and technology for decision making (Matt et al. 2015). Finally, since the list could be very long and related to what was expressed at the beginning of this section, authors such as O'Reilly focus on the journey of no return that any global digitalization process entails (O'Reilly 2017).

Logically, as a conclusion to this first introductory section, we can confirm the relevance of the terms digital transformation and artificial intelligence and their importance in the field of business. However, it is important not to forget the dangers that this new technology entails and the need for clear regulation to guarantee that its use respects the fundamental rights of citizens. In this sense, work will be carried out in the following sections, giving special importance to the ethical aspects derived from this new reality, analyzing its problems and threats while reviewing the different legal frameworks.

## 2. Literature Review

As we pointed out in the previous section, digital transformation and artificial intelligence are undoubtedly changing business models and improving communication between people, which in turn facilitates business structures. This is a truly global business revolution in which technology has brought about major changes in all processes, even going so far as to digitize the products themselves. There is no doubt that technology is the key enabler of digital transformation, and artificial intelligence is its main player.

In the digital transformation process, the most relevant thing at a global level is the transformation of the business itself, but this process should be understood not as an end in itself but as a means to increase the efficiency and sustainability of the business. Thus, Vacas Aguilar points out that "digital transformation constitutes the great pending process to be completed in a large majority of organizations after a first stage of integration of digital devices and networks" (Vacas Aguilar 2018). The success or failure of this process will ultimately depend on the definition of clear and quantifiable objectives of a digital strategy aimed at achieving these objectives, setting the tactics and key variables to be followed in the process (Baker 2014). The concept of digital ecosystem encompasses, in fact, "a collective of companies interconnected by a common interest in the prosperity of a digital technology to materialize their own innovative product or service" (Selander et al. 2013). Other authors define this digital ecosystem not only from the point of view of interconnected companies but more broadly by relating it to "the environment in which digital objects are embedded in changing interdependencies with other entities" (Kallinikos

et al. 2013), while other scholars emphasize technological interconnectedness by referring to digital ecosystems as sets of information technologies that are related based on a specific context of use (Adomavicius et al. 2008).

As we have said, at this point we will also briefly review the academic contributions that have defined the precedents to this so-called fourth industrial revolution. Thus, the digital revolution is in the midst of this industrial revolution and is driven by the convergence of computing, artificial intelligence, universal connectivity and data. There has been a massive acceleration of technological evolution itself "driven by a massive expansion of our ability to store, process and communicate information using electronic devices" (Eurofound et al. 2018). From work to leisure, digital technology is undoubtedly changing our habits. Obviously, this also affects the way we do business and even the way we cooperate and interact with others. Digital communications, social media interaction, e-commerce and digital business are constantly changing our world. The generation of data is exponential and the new possibilities for its management and use have also brought new value. These changes are brought about by the industrial and digital revolution (European Commission 2020a). The fourth industrial revolution has a combined effect of innovations such as artificial intelligence, robotics, blockchain, etc., that puts us on the brink of a major change in the way we live and work, giving rise to the so-called digital disruption (Hinings et al. 2018).

The term artificial intelligence applies to systems manifesting intelligent behavior capable of analyzing their environment and taking action, with a certain degree of autonomy, in order to achieve specific objectives (Butterworth 2018). The Resolution of the European Parliament of 16 February 2017 already points out recommendations addressed to the European Commission on civil law rules applied to artificial intelligence (software, hardware, robotics, etc.) (European Parliament 2017).

Based directly on the global effects of artificial intelligence, authors such as Russell and Norvig, group the definitions around two axes (Russell and Norvig 2020):

- Axis 1: Thought—Behavior.
- Axis 2: Human—Rational

This double group would be the typical approach of the Turing Test (French 2000). The computer will pass the test if faced with an interrogation by a human, the human cannot determine whether the answers were provided by a computer or another person. In Russell and Norvig's own orientation of AI towards human behavior, the computer must possess the following skills:

- Natural language processing to enable it to communicate.
- Knowledge representation to store what it knows or what it hears.
- Automatic reasoning to use the stored information to answer questions and infer new conclusions.
- Automatic learning to adapt to new circumstances and detect and extrapolate patterns.

In this way, and as we will see later, artificial intelligence approaches must take into account human behavior, human thinking, rational thinking and rational behavior (Pedreño Muñoz 2017).

AI therefore has great transformation potential from technological, economic, environmental and social points of view given its intersectoral penetration, high impact, rapid growth and contribution to improving competitiveness.

All of the above justifies the convenience of implementing national strategies for artificial intelligence that allow structuring the digital transformation action of the different administrations and provide a frame of reference and momentum for the public and private sector.

For example, Spain has recently developed a National Artificial Intelligence Strategy (Ministerio de Asuntos Económicos y Transformación Digital 2020b) that responds to the shared commitment with its European partners for the EU to become a leader in this area. This commitment is included in the Digital Agenda for Europe (European

Parliament 2020), the Strategy "AI for Europe" adopted in 2018 (European Commission 2018), "How to measure digital transformation" of the OCDE (OCDE 2019), the "White Paper on Artificial Intelligence" published in February of 2020 (European Commission 2020b) and the European policy "Artificial Intelligence (European Commission 2021).

The coverage of legal and human rights issues is also wide. International coverage of legal and human rights issues is evident and comprehensive in policy documents of the United Nations (United Nations 2019); OECD (OECD 2019); Council of Europe (Comisión Europea 2017, 2018, 2019), the European Parliament (European Parliament 2020), European Union 2021, the European Commission (2018, 2020a, 2020b), European Commission for the efficiency of justice (CEPEJ 2018) and the European Data Protection Supervisor (European Data Protection Supervisor 2016).

Academic and civil society coverage of AI-related legal issues is also broad (Access Now 2018; Privacy International 2018) and covers a variety of risks and issues yet to be resolved. Some of these include considerations as broad as those related to intellectual property (Schönberger 2018), privacy and data protection (Mittelstadt et al. 2016), the workplace and its impact (De Stefano 2019), opaque algorithms (Lepri et al. 2018; Coglianese and Lehr 2018) or damage management and associated responsibilities (Vladeck 2014).

Given that, as stated in the introduction, the aim of the article is to analyze the ethical and legal implications from an academic perspective, it is necessary for this literature review section to pay attention to these aspects. Recently, in April 2021, the European Commission published the basis of the regulation on the use of artificial intelligence. The aim of this new regulation on AI is to ensure that citizens (in this case, Europeans) can have confidence in what AI can offer, to guarantee the protection of fundamental rights against the risks that may arise from the use of tools or systems based on artificial intelligence. This is important to note as it is the first-ever legal framework on AI and a new plan coordinated with the member states. The objectives set out in this regulation aim to ensure the safety and fundamental rights of individuals and businesses, while at the same time aiming to strengthen investment and innovation in AI across the EU. In the same regulation, new rules on machinery using artificial intelligence have been developed (PriceWaterhouse 2021). Continuing with this same regulatory proposal, authors such as De Miguel Asensio point out that to a large extent, the proposal has been built on the model of pre-existing legislation on product security and is expected to have an impact and influence in other parts of the world as happened with regulation 2016/679 on data protection (De Miguel Asensio 2021).

In the same sense, the aforementioned proposal has gone in parallel with the European Commission's intention to revise the Directive on Product Liability to adapt it to the requirements of the new technologies inherent to AI (European Commission 2021).

In short, the ultimate aim of this literature review is to consider the relationship between the importance and validity of the processes of digital transformation and artificial intelligence in the business sphere with their ethical and regulatory components. In any case, compliance with new obligations should not be interpreted as an obstacle to the digital transformation of the different economic sectors. On the contrary, the proposed AI regulation establishes a stable framework that will protect the uses of AI and facilitate the interpretation of diligent conduct, which is essential for the internal and external risk management of companies. In other words, the new regulatory framework is an opportunity to develop new AI applications with legal certainty, combining innovation with high-security standards, all in favor of user and consumer rights (Martínez Moriel 2021).

## 3. Materials and Methods

The above approaches to AI determine different lines of work and algorithmic learning models. However, they all converge in the use of cognitive technologies enabling a fundamental change in the way we interact with machines (Blank 2017). Just as people use five primary senses (sight, hearing, smell, taste and touch) to relate to our environment, we

can also think of the "senses" of AI systems that enable them to hear/speak, see, remember (knowledge) and analyze/act.

The technologies associated with these five AI senses are the so-called core components of AI in that they are the building blocks from which applications are built or use cases are developed. Any enterprise AI platform or system can be divided into solutions based on one or more of these core components.

Speech and hearing: Virtual Personal Assistants (VPAs) such as Apple's Siri, Google's Google Assistant, Microsoft's Cortana, Amazon's Alexa or Baidu's Duer, already allow users to search for information and execute commands from voice inputs (Yu and Deng 2016). This is the rise and growth of PLN (Natural Language Processing) platforms (Nadkarni et al. 2011). Linked directly to the business arena, chatbots are being deployed to improve productivity and efficiency in customer service (Dale 2016).

Any conversational agent currently (Virtual Personal Assistant or VPA) is composed of the following elements:

- User interface to accept inputs (voice or text commands).
- NLP (Natural Language Processing) or speech recognition element to understand user inputs and manage the dialog by contextualizing the conversation.
- Back-end infrastructure that connects the bots/VPAs to the different applications/services (Angga et al. 2015).

Apart from B2C (Business to Consumer) applications, companies are also pushing the implementation of conversational bots in internal process support functions such as supplier payments/billing, HR recruitment and training, and administrative processes.

View: Computer vision or machine vision is the ability of machines to extract meaningful information from images or videos. Image and video traffic is presently the major cause of the growth of so-called "unstructured" data (Kaur and Sood 2017). This is a challenge when it comes to manual visualization and, above all, when it comes to tagging and indexing this type of information for subsequent analysis.

Machine vision makes it possible to host a new generation of applications (Wang et al. 2019b). Advances in machine learning techniques (together with improvements in camera hardware and computing capacity for processing) have led companies to adopt these technologies for tasks such as the identification of objects, people, facial expressions, activity monitoring and video surveillance (Wang et al. 2019a). Machine vision is also generating a wave of innovation in the field of robotics, extending its use cases across multiple sectoral approaches. Examples include medicine (image analysis for early disease detection (Khemasuwan et al. 2020)), retail (automatic stock control, facial recognition to improve the shopping experience (Xu et al. 2020)), automotive (assisted driving systems . . . ), assembly lines (assembly line monitoring) and security (surveillance (Rashmi and Rangarajan 2018)).

Remember: Data discovery and the ability to integrate data are important characteristics of AI systems. Their correspondence in humans is the ability to store information and experiences and to be able to recall and retrieve them later. Without access to memory and the ability to retrieve information, the ability of an AI system to locate, process and act on information is severely restricted. Understanding data relationships, their structures and storing these relationships, as the brain encodes relevant information in what we call 'memory', is an iterative and dynamic process (Garnelo and Shanahan 2019). Data integration tools play an important role in the processes of storing, searching and mining data as a preliminary step before effectively applying machine learning techniques (Nguyen et al. 2019).

Analyze and act: A key feature of AI-based models and applications is their ability to translate raw data into actionable information. This is one of the reasons why companies are deploying AI-based tools, thus facilitating decision-making (Collins and Moons 2019). In this sense, AI solutions go beyond the descriptive and diagnostic functions of traditional analytical tools and address predictive and prescriptive aspects. That is, they not only explain what has happened but also predict what may happen and prescribe action. Additionally, these solutions can be deployed by developing both in-house on-premise

capabilities and with cloud or hybrid solutions (Gill et al. 2019). The consulting firm Gartner has estimated that the market for data science platforms has grown from \$2B in 2016 to \$5B in 2021, with 15% CAGR growth over that period (Baker et al. 2020). The Figure 4 shows the growth potential of different AI-related technologies.

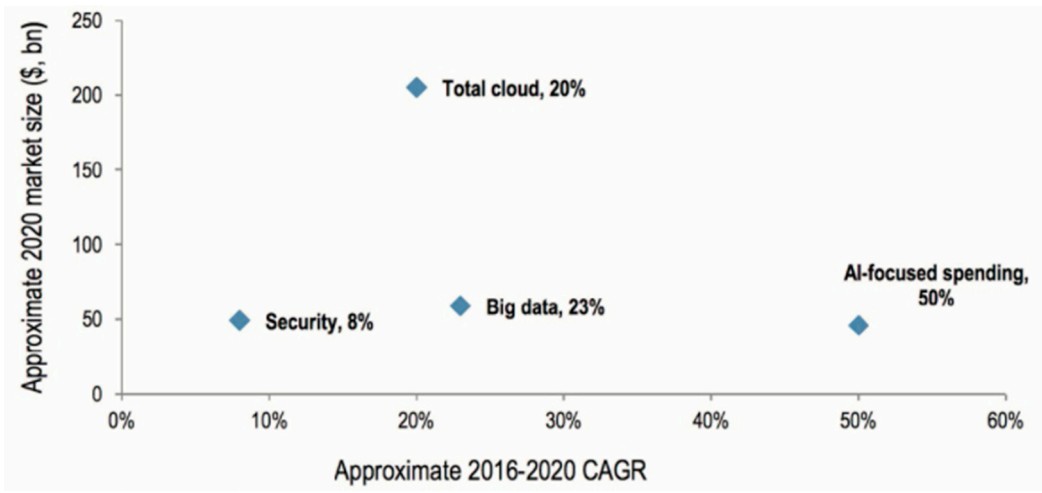

**Figure 4.** AI market growth compared to other high-growth technology segments. Source: (Gartner 2020).

### 3.1. Methodology

In this emergence of new disruptive technologies that have defined the process of digitization and industrial transformation itself, we can find:

- On the one hand, the technologies that address the physical and that are related to aspects such as biotechnology, robotics, the Internet of Things, 3D printing, new ways of using energy and many others.
- On the other hand, those technologies that have more to do with digital, such as Blockchain, new computational capabilities, Big Data, virtual reality, augmented reality and, more globally and holistically, artificial intelligence.

The four key points that are going to mark this disruption are:

- Data storage and management capacity: storage capacity of a high volume of data thanks to Cloud technologies and its management and processing based on the use of Big Data technologies.
- The processing power of this information: high volumes of data and normally unstructured require high computing capacity.
- Improved communications: enabling access to data in the cloud with high speed and minimal latency.
- Advances in mobility and different access points: making it possible to access data wherever it is generated.

In order to methodologically develop this whole process, the concept of machine learning will be fundamental, defined as a subset of Artificial Intelligence aimed at recognizing data patterns and making predictions, partially based on the principles of classical statistics. These advanced data processing capabilities allow machine learning-based models to perform different types of analysis:

- Descriptive analysis: describes what has happened. It is widely used at the enterprise level because of its simplicity.
- Predictive analytics: anticipates what will happen and is mainly based on probabilistic techniques. It is often used in data-driven organizations as an element in decision-making.

- Prescriptive analytics: provides recommendations on what to do to achieve an objective. It is used in companies with a high degree of digitalization because it requires large volumes of data.

All companies are flooded with data. The real need lies in those capabilities that provide "intelligence" to the data—the so-called "smart data". Organizations need to process these data, as well as coordinate activities both inside and outside the organization to the results achieved from data analysis (Ross 2014).

Machine learning, as was the case when defining artificial intelligence, is not only about analyzing but also about learning. Thus, there are three main learning methods based on machine learning:

- Supervised learning: the algorithm uses training data and feedback provided by humans to learn the relationships between input and output data. With this, the algorithm determines the logic that can be used to predict the response. This method is used when we know how to label the input data and the type of behavior we want to predict, but we need the algorithm to calculate it automatically with new input data. The algorithm (linear regression, decision trees, Naive Bayes, Random Forest, AdaBoost, Affinity Analysis, etc. (Witten et al. 2005)) is trained with the labeled data to find the connection between the input variables and the result. Once the training is completed, usually when the algorithm is already sufficiently accurate, the obtained model is applied to new data. Some use cases of supervised learning methods can be applied to different fields such as: predicting call volume of a call center for sizing purposes; detecting fraudulent activity in credit card transactions; predicting the demand for a product and the necessary inventory levels; predicting the probability of a patient joining a health program, etc.
- Unsupervised learning: in unsupervised learning, the algorithm explores the input data but without being explicitly provided with an output variable or response. Unsupervised learning is conceptually modeled in the same way that humans observe the world: drawing inferences and grouping things based on observation and intuition. As our experience increases (or in the case of machines, the number of data being processed grows), our intuition and observations change or become more refined. This method is used when you do not know how to classify the data and want an algorithm that finds patterns and classifies the data for us. The algorithm (K-means clustering, Gaussian Models, hierarchical trees, etc.) receives unlabeled input data and infers a structure from those data, identifying groups of data that have similar behavior. Some use cases of unsupervised learning methods are the following: segmenting customers into groups with different characteristics to optimize the performance of marketing campaigns; recommending movies to users based on the preferences of customers with similar attributes; recommending new books based on previously purchased books, etc.
- Reinforcement learning: in reinforcement learning, the model is provided with a set of allowed actions, rules and potential end states. In other words, the rules of the game are defined. By applying these rules, exploring different actions and observing the resulting actions, the machine learns to use the rules to maximize the outcome. That is, the algorithm learns to perform a task by trying to maximize the reward it receives for its actions (Barto and Sutton 1997). Reinforcement-based learning is equivalent to teaching someone to play a game. The rules are defined, but the outcome varies according to the judgment of the player, who must adjust to the context of the game, his skill and the actions of the opponent. This method is used when there are not many data to train the algorithm, and the ideal state cannot be defined. The only way to learn about the context is to interact with it. The algorithm takes action (for example, buying or selling stocks) and receives a reward if the action brings it closer to the goal of maximizing the total possible rewards (for example, doubling the value of the stock portfolio). The algorithm optimizes the outcome by correcting itself all the time to achieve the best possible set of actions. Some use cases of reinforcement learning

methods are optimizing trading strategies, stock management, etc.; optimizing the behavior of autonomous cars; optimizing prices in real-time online based on products with low stock or foreseeable variations due to competitor campaigns, etc.

As can be seen in Figure 5, machine learning will be one of the predominant technologies in the next decade (2020–2030).

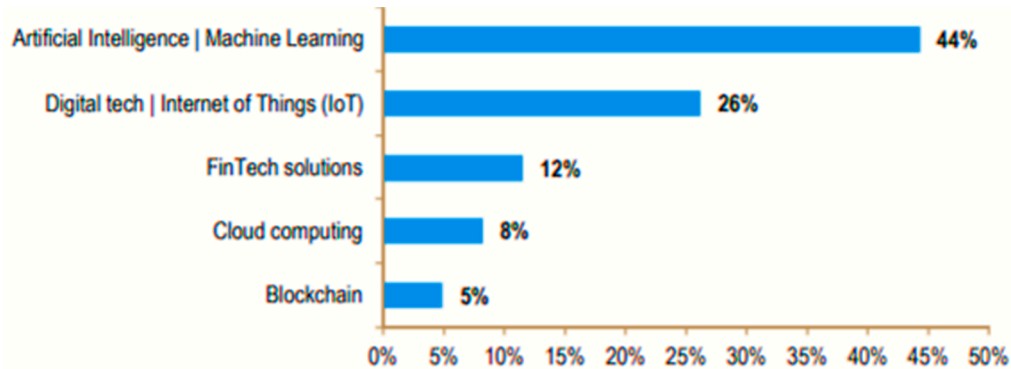

**Figure 5.** Technologies having the greatest impact in 2020–2030. Source: Own elaboration based on (Gartner 2020).

In short, the application of technology to the business environment considers innovation as the basis for radically improving the performance or scope of companies (author, year).

In general, as authors such as Fitzgerald point out, when all hierarchies and business processes are influenced at the same time, the new reality often leads to completely new business models (Fitzgerald et al. 2013). The described idea of digital transformation under the umbrella of artificial intelligence and machine learning shows that its multifaceted nature surpasses the level of any past transformation or innovation. This reality becomes clear when observing the real difficulties in all industries, almost without exception, when it comes to carrying out digital transformation processes (Davenport 2020). Indeed, although the theory is often well known, all organizations are aware of its fundamental importance and still face numerous obstacles that prevent them from initiating, let alone progressing towards, a complete digital transformation process.

Thus, as various authors have pointed out, only a notable minority of organizations have prevailed in terms of creating the right administrative and mechanical skills to reap the transformational impacts of new digital innovations (Cruz-Jesus et al. 2017). In other words, this new dimension is not just about launching new products or services to the market but must adapt all internal and administrative processes to a new digital dimension (Parviainen et al. 2017). These implementation difficulties are given by the need to coordinate aspects such as Big Data, computational calculation, Cloud solutions and programs based on algorithm development (McKinsey Global Institute 2011).

### 3.2. Biases and Explainability

What was a few years ago an ethical issue, avoiding discrimination when using data and algorithms in decision making, has now become a legal issue. In a technological and scientific environment with such rapid and unpredictable evolution, the regulatory environment is not always able to arrive in time to protect the rights of citizens, so the specialist, scientists and technicians must maintain high ethical standards in the day-to-day of their work.

There are a number of points to keep in mind in this regard:

- Always keep in mind the data protection regulations.
- Make sure that the algorithms we use do not involve decision making that implies the discrimination of any group based on age, sex, race, religion or any other aspect.
- Check that the data used do not contain bias that could lead to wrong decisions.

- Interpret the results of the models scientifically, avoiding interpretations interested and not adjusted to reality.
- Use the appropriate working methods that guarantee the reliability of the results.

Concepts such as artificial intelligence and machine learning are part of the daily vocabulary of works councils and governments around the world. Decisions made based on these models affect millions of people. Therefore, it is the responsibility of the experts to ensure that the final objective is to provide benefit to society in general or the clients of the company in particular while respecting the fundamental rights of citizens.

### 3.2.1. Biases: The Reason Why Algorithms Learn and Their Main Weak Point

Biases are impossible to eliminate since it is the mechanism by which algorithms learn. Creating biases and making global assumptions while neglecting concrete details is the basis of learning, and, therefore, exceptions are always going to be there. The important thing, in this case, is to minimize them and that these unlearned biases are not training-induced biases due to a poor choice of training data or age, sex, race or religion issues.

It is therefore very important to take these types of issues into account in sensitive environments with biases that can incur discrimination of any kind. If an individual should not be discriminated against for ideological, gender, ethnic or other reasons, algorithms should not do so either.

To avoid biases, we should take into account the following points:

- Select training data carefully;
- Validate the algorithms not only with data from the first world or our area of influence but also from other parts of the world with different traits, cultures or ethics;
- Maintain continuous surveillance of the decisions they make, to intervene as soon as possible if these biases are detected;
- Have human evaluators who confirm the decisions made by the algorithms or at least that the users who are affected by them can come to have their particular case reviewed.

This is going to be one of the most important challenges in a few years when AI algorithms are the ones that control more and more processes in which the lives of citizens are affected.

Regarding the responsibility for the errors of these algorithms, it is quite difficult to establish it, and it is something to which we must strive as a society. Obviously, the responsibility should always be on the organization using these algorithms to test them correctly. However, nothing is 100% error-free. Therefore, it must be established what the allowable margin of error is for each task that is delegated to the AI. After all, humans are also wrong. The difference here is that the chain of responsibility between humans is traceable and more or less clear. When an AI algorithm comes in, it is not so clear anymore: is it from the algorithm designers who selected the training data or someone who did not correctly review the algorithm's decision? This is an important challenge to consider in the future.

### 3.2.2. Algorithm Explainability

If we can explain the model that is generated in the learning process, it is easier to avoid or predict these errors. However, this is not always possible because not all algorithms are explainable or easily explainable. The explicability gives us a series of issues that we must take into account, such as:

- Reliability: It is very important to be able to trust the decisions of an algorithm, especially if it is in charge of something important. For example, if it is driving a vehicle, making purchasing decisions on the stock market or operating a nuclear power plant. However, also for other minor issues, it is important to know what decision the algorithm is commanding because it may incur biases, discrimination,

etc. Knowing of how the decision is arrived at can help us prevent this bad behavior of the algorithms.

- Acquire new knowledge: Algorithms are sometimes capable of solving problems or discovering new solutions to problems that were not known before. However, these problems often cannot be analyzed correctly because we do not know how the algorithm came to that conclusion. Therefore, we lose the details of that newly acquired knowledge.
- Failure detection: If the model has failures and we know the model, we can predict, mitigate or retrain them. So far, we can only tell if a black box algorithm is flawed by testing it thoroughly. However, there can always be cases that you have not contemplated in which the algorithm fails.

In summary, we must be aware of domains for which the model's lack of explainability is not a problem and in which benefits of these models compensate for the lack of explicability, and in which domains it is advisable to use explainable algorithms, even if they obtain worse performance.

### 3.3. Ethical Use of Artificial Intelligence

Additionally, artificial intelligence faces one of its greatest challenges, which is its responsible and ethical use, also within companies. According to the (Boston Consulting Group Report 2021), 55% of companies overestimate the maturity of their responsible Artificial Intelligence initiatives. These programs are structured around three axes: justice and equity, mitigation of social and environmental impact, and human or ethical AI.

These Basic Principles of Artificial Intelligence have a further development from the AI Principles of the Asilomar conference (AI Principles 2017), which structures them in three blocks:

- Research problems. The goal of AI is to create intelligence that provides direct benefit, with a constructive and healthy exchange between AI research fields and policymakers, fostering a culture of cooperation, trust and transparency between researchers and developers of IA, and with investments that guarantee that there are no cuts in safety regulations.
- Ethics and values. AI systems must be safe and protected throughout their operational life, allowing transparent analysis of their operation and in a verifiable way when applicable and feasible. Highly autonomous AI systems must be designed, ensuring that their goals and behaviors align with human values, and people must have the right to access, manage and control the data that are generated. The profit and prosperity generated by AI technologies should be targeted to as many people as possible, avoiding an AI-led arms race.
- Long-term problems. The profound change that AI can represent, especially those catastrophic or existential risks, and applying strict security and control measures, must be planned and managed with the appropriate resources. Superintelligence must be developed for the benefit of all humanity rather than a single state or organization.

The action of the European Union in the field of Artificial Intelligence has been of high intensity in the last five years, including in the regulatory field (Comisión Europea 2021), highlighting, for example, the White Paper on Artificial Intelligence of February 2020 (Comisión Europea 2020). The European Parliament also issued a Resolution on intellectual property and AI (European Parliament 2020). In April 2021, the European Commission launched a proposal for an Artificial Intelligence Law (European Union 2021), aimed at regulating the prohibited uses of AI and especially those called high-risk for which compliance with strong requirements is required in various areas: use and data quality, its governance and transparency, the need for supervision by people, cybersecurity, etc. This regulation does not regulate key issues such as liability or intellectual property or other forms of protection of algorithms.

Specifically, Section 2 of the present paper establishes a list of AIs that are prohibited because they imply an unacceptable risk for contravening the values of the Union, for

example, for violating fundamental rights. Section 3 of the paper develops what are called high-risk AI systems. It is the core of regulation and defines them as those that involve significant risks to the health, safety or fundamental rights of people. It guides the minimum requirements necessary to address risks and tries not to limit or hinder technological development and avoid increasing the cost of marketing AI-based solutions. It is especially relevant in the legal requirements, article 10 "Data and Data Governance" and especially Section 5, where it tries to harmonize this regulation with the RGPD (General Data Protection Regulation).

In summary, the risk-based approach of this regulation is considered entirely correct to combine the legal certainty of providers and developers with that of users. It would also be appropriate to include a regulation of specific uses to increase the precision and efficiency of the risks.

## 4. Results

This section will try to highlight the results of the literature review in relation to the importance of digital transformation and artificial intelligence linked to business.

There is no doubt that artificial intelligence has made its way into reality, not only economically, but also in people's daily lives (Reier Forradellas et al. 2021a). The magnitude of the challenge is of global stature, a reason that is leading global authorities, including the EU, to which reference has already been made, to address the legal issues involved. This fact makes it possible to raise, in a concrete manner, the challenges that arise and the areas that might need uniform legal treatment, as a means of contributing to the development of systematic and international solutions, since traditional methods of regulation are not fully applicable.

Recent public debates have revolved especially around the need to regulate the AI sphere itself and to set limits, to prevent the development of so-called artificial general intelligence, i.e., an intelligent system comparable to or even superior to human intellectual capacity. In addition, discussions point to the need to teach ethics to artificial intelligence systems and to include in them the values that society recognizes.

This regulatory need becomes even more necessary if one takes into account the volume of business that Artificial Intelligence will generate and already generates. To put this global process into figures, according to the consulting firm McKinsey, the acceleration of digitalization could mean an extra business volume of two trillion euros in Europe over the next decade. In the case of Spain, the same estimate puts this impact at 1.8% of the total Gross Domestic Product by 2025 (McKinsey 2020). The Figure 6 shows the increase in online sales in Spain.

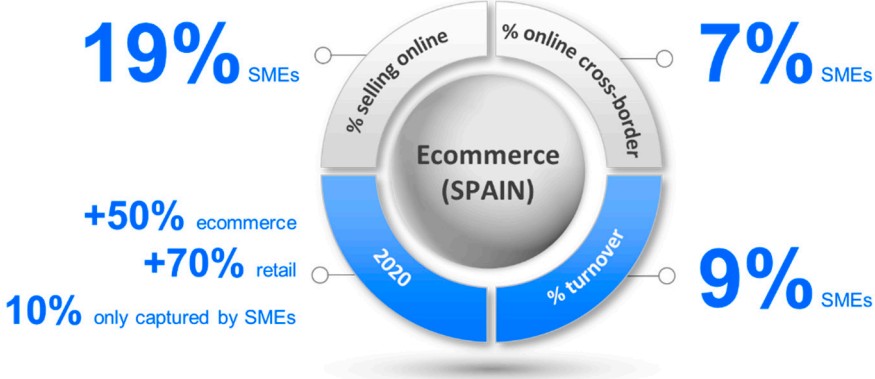

**Figure 6.** Increase in online sales in Spain, 2020. Source: Own elaboration.

This digital transformation for the coming years is so important that in the Spanish Digital Agenda (Ministerio de Asuntos Económicos y Transformación Digital 2020a, 2020b),

it is already included as an inalienable objective that in 2025 SMEs capture 25% of the growth of the online business (currently they do not reach 10%).

More precise recognition of the influence of digital transformations involves two organizational aspects:

- An impact referred to the outside of the organization. From this dimension, there is an improvement in the customer experience and a change in the entire process of the customer–company relationship, from the first commercial action to the post-sales service itself.
- An impact referred to the interior of the organization. This impact directly affects the structure and functioning of organizations. The impact on business objectives, on new labor and leadership relationships and on hierarchical structures has led to a new dimension of all organizations. This new dimension has a key aspect: it is an obligation and not an option. In other words, those organizations that do not know how to adapt to this new environment will find it very difficult to survive.

According to forecasts, the use of AI-based learning models will increase significantly in the coming years. In 2023, 40% of development teams will use machine learning-based services to build models that incorporate AI capabilities into their applications, compared to 2% in 2019 (Boston Consulting Group Report 2021). By 2025, 50% of the activities performed by data scientist profiles will be automated using AI. (Baker et al. 2020).

This increase in the use of AI-based services is reflected in the market estimates made in the IDC study (2021), which indicates that the worldwide profits of the market for Inteligen and AI solutions, including software, hardware and services, are estimated in 2021 at $327.5 billion, with an annual growth of 16.4%. In 2021, the market exceeds $500 billion with a five-year compound annual growth rate (CAGR) of 17.5%.

Therefore, this expansion of AI-based technologies implies that AI spending will double in four years. The components that make up this spending are:

- AI applications: applications that learn, discover and make recommendations/predictions or AI building blocks;
- AI software platforms: tools built on top of AI building blocks that enable AI-based use cases;
- AI professional services: consulting and implementation services for AI technologies provided to enterprises;
- AI hardware;
- AI computing and storage capacity.

Artificial intelligence can transform the productivity of companies and is far-reaching enough to influence the GDP of the global economy (Alcaide and Díez 2019). According to the PWC study (2017), the largest economic gains from AI will occur in China (26% increase in OIB by 2030) and the US (14.5% increase), which is equivalent to a total of $10.7 trillion and represents almost 70% of the global economic impact. Following on from studies already noted in the present paper (McKinsey 2020), based on surveys of over 2300 participants in June 2020, 22% of respondents reported that at least 5% of their companies' EBIT was already attributable to the impact of AI.

The continued erosion in companies' profitability is a risk that conditions investment and the very value of their shares. However, artificial intelligence is already seen as a new productive element that contributes to lowering this risk, with a potential to raise profitability rates by an average of 38 percentage points, implying an economic boost of $14 trillion in GVA (gross value added) by 2035 (Accenture 2017). The sectors with the highest potential impact (as shown in Figure 7), above USD 300B, are:

- Industry (manufacturing): USD2071B;
- Sales (wholesale and retail): USD 943B;
- Professional Services: USD 569B;
- Information Technology and Communications: USD 527B;
- Financial services: USD 461B;

- Transportation and warehousing: USD 300B.

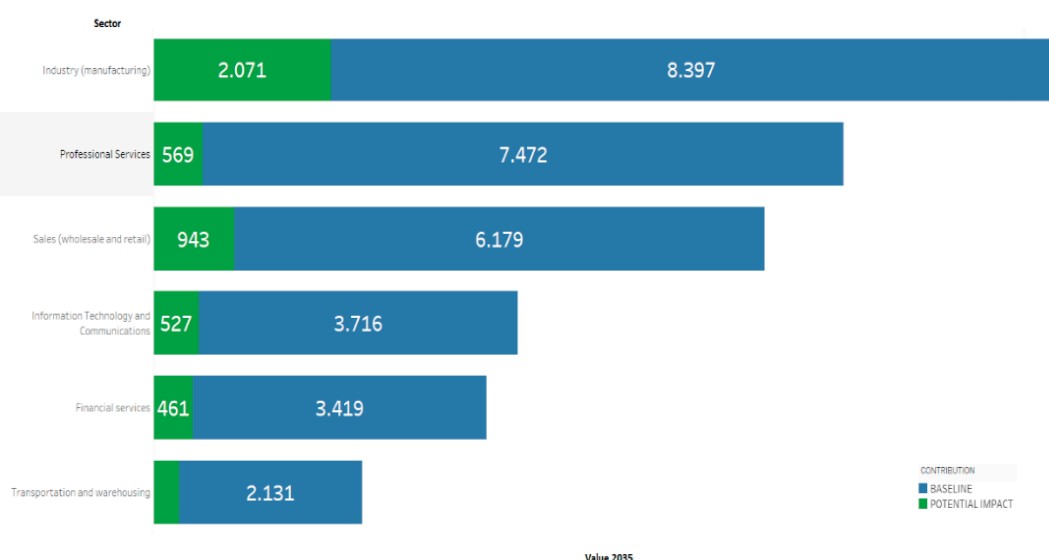

**Figure 7.** Baseline GVA with AI potential impact added in 2035 (US$B). Source: Own elaboration based on (Accenture 2017).

Logically, the data taken into consideration explain the need to promote the use of these technologies in the business sphere (Reier Forradellas et al. 2021b), but as has already been pointed out in this paper, their use must be accompanied by a regulation that guarantees concepts of security and non-infringement of rights. Thus, the Proposal for an AI Regulation already mentioned in this paper (and which may still undergo modifications) establishes administrative fines of up to 6% of annual global turnover or up to 30 million euros for those companies that fail to comply with their obligations derived from the use of AI (Martínez Moriel 2021). As a final consideration in this section on the results, it is necessary to question the figure of the authorities in charge of regulatory control. It is worth warning of the risk they may run if they do not receive the necessary funding and legal powers to carry out their functions, including the capacity to assume the workload that the regulated activity requires. The establishment of a sanctioning regime is a necessary but not sufficient condition to ensure regulatory compliance, as has been shown on many occasions in the framework of personal data protection (Zuiderveen 2020). Governance mechanisms need to be defined to enforce the application of ethical principles so that they are transformed into legal obligations for AI technologies (Mittelstadt 2019).

It is necessary to clarify the direct and indirect roles that ethics can play in the regulation of technology (López de Mantarás 2017) by expanding the object of analysis to include its use and moral implications for the social context (Gurkaynak et al. 2016). On the other hand, the discussion should also address the issue of labor relations themselves as improvements may entail the replacement of thousands of jobs. Finally, it is convenient to emphasize the need to establish additional considerations to responsibility (Bertolini 2014), not only at the individual or corporate level but also at the national and international level, since the use of new technologies may go beyond individual responsibilities.

As a conclusion to this section, it seems reasonable to raise a number of methodological and substantive questions about the regulation of technology in general and AI in particular. First, it is worth asking whether sufficient arguments can be identified to redefine these new technologies, and thus justify a change in the legal framework. That is, are existing laws sufficient to meet the regulatory challenges of the technology, and if not, should they be adapted to include the new technology?

The following section devoted to conclusions will answer the previous question and propose regulatory solutions in this regard.

## 5. Conclusions and Discussion

As this paper has tried to show, digital transformation is no longer a buzzword, it is a fact. Almost 70% of companies are aware of the benefits of digital transformation for their business models and the need to develop strategies linked to digitization. Digital transformation refers globally to organizations and companies of different shapes and sizes, from different sectors and with different objectives and needs but with a similarity, in all cases, that it is the incorporation of new disruptive technologies that mark a turning point when defining new business models.

Similarly, artificial intelligence is not just a fashionable concept. Companies are deploying artificial intelligence to address improvements in customer experience, strengthen the search for efficiencies and even generate new business models.

According to the McKinsey study (2020), artificial intelligence has a higher level of adoption in product and service development and in functions associated with the operation of services. This is followed by marketing- and sales-oriented use cases, such as customer service analytics or customer segmentation and cases related to fraud management and risk modeling. Finally, there are manufacturing, HR, logistics and corporate finance environments.

Therefore, AI not only generates the possibility of automating functions and, in this way, freeing up time and effort, but when applied in a more creative way, it also makes it possible to identify new opportunities, new products, and new ways of reaching customers with new channels.

In the medium and long term, AI will be an intrinsic part of many applications and will therefore be universally deployed, but we can affirm that both companies that have already been involved in the development of these technologies at an early stage ("early adopters") and those that adopt and implement them quickly ("fast movers") will be able to capitalize to a greater extent on their advantages. Likewise, in the opposite direction, agents that do not consider their adoption will be less competitive.

Computers are now capable of performing tasks that were previously considered the exclusive domain of the human mind. These self-learning systems are affecting almost every vertical sector, from manufacturing to financial services, giving rise to new business models while making some legacy models obsolete. With their application, companies are mainly seeking the following benefits:

- Drive sales and customer engagement. AI-assisted marketing platforms can automate digital marketing and target high-value customers, for example, when launching a new product, to identify the characteristics of high-value customers and the products they purchased or target customers most likely to buy new products. AI makes it possible to target all customer segments in a fully personalized way. AI can enhance the customer experience in a multichannel world. Applications include recommender systems, virtual assistants, chatbots and voice bots. Virtual assistants or agents can handle higher volumes of customer service interactions, especially if they are repetitive or routine tasks, thereby increasing customer satisfaction.
- Promote operational efficiency. AI capabilities are improving quality control and predictive maintenance in industrial environments. Efficiencies range from reduced operating costs to improved machine and process performance. In other administrative areas, AI can automate processes that require the processing of large numbers of data and may include variations in the input information. This is the case with the processing of customer orders or other recurring tasks.
- Improve products. The incorporation of AI in the product or service itself (such as Movistar Aura, Telefónica's virtual assistant in Smart Home) can improve interaction with the customer or simply boost the product's functionalities in an advanced way.
- Generate new relevant information or develop new business models. Better data analysis is enabling companies to think differently and more creatively. Employees can spend less time on routine tasks, reduce human error and think of new products and new go-to-market strategies by developing a deeper understanding of customers.

In short, companies that combine strong digital capabilities with robust adoption of AI technologies and a proactive AI strategy will have above-average financial performance. Technology is a tool and, on its own, does not deliver productivity improvements. It needs to be accompanied by the identification of relevant use cases, the development of internal capabilities, and change management that generates agile work environments and collaborative culture.

In this context, although the technological barriers are being overcome (although the speed at which they are evolving is also a challenge today), there are still significant challenges to the adoption of AI in companies. These factors have to do with the insufficient existence of specific AI talent (data scientists, machine learning experts, etc.) and the usual difficulties in demonstrating the value of technologies that are not yet mature enough, such as quantifying the value in the deployment of AI solutions and the lack of strategic impetus from the management levels of companies.

Additionally, artificial intelligence faces one of its greatest challenges, which is its responsible and ethical use, also within companies. According to the Boston Consulting Group report (2021), 55% of companies overestimate the maturity of their responsible artificial intelligence initiatives. These programs are structured around three axes: justice and equity, mitigation of social and environmental impact, and human or ethical AI.

In this context, it is essential to generate legislation that takes into account the interests of all stakeholders (company, consumer, global market, national countries) to provide guarantees that these digital transformation processes meet a certain regulatory standard. The EU is committed to a reliable AI based on regulatory and ethical compliance. In April 2021, the commission launched a proposal oriented to this objective (Artificial Intelligence Act) that regulates the uses of AI and prohibited and high risk scenarios, including guidance around the data. In this respect, situations where a choice between business outcomes and regulation will be common. Taking into account what this paper has provided, it seems clear that artificial intelligence should never come before the basic rights of citizens. In these cases, as established in the regulation itself approved by the European Union, an acceptable level of risk must be considered for the development of these technologies:

- Inadmissible risk. In these cases, systems linked to AI that are considered a clear threat to the security and rights of individuals should be banned outright.
- High risk. In these cases, the potential for security risk and infringement of rights must be analyzed. All such AI-linked systems that are considered potentially high-risk will have to have strict compliance requirements before being granted marketing authorization.
- Limited risk. Providers of these so-called limited-risk systems should be required to comply with specific transparency obligations to ensure that users are directly aware of their compliance.
- Minimal or no risk. In these cases, the presence of a regulator would not be necessary as this group of systems would have minimal or no risk in terms of security and infringement of rights.

Consequently, it seems clear that the sector needs guidelines or regulations so that when they undertake these processes, they do so with a certain guarantee. With regard to the message that the public administration should convey in terms of digital market legislation, a regulatory framework should be considered to facilitate and guide companies that have not yet achieved their digital transformation so that they can do so. There should be minimum guidelines to homogenize the business fabric in order to maintain competitiveness in the global market. However, these rules must be clear and enforceable. Precisely on the basis of this situation, another question arises for future work. Once the regulatory needs have been defined, it will be necessary to define which artificial intelligence applications fall into each of the categories mentioned above. There are, logically, cases of easy application. Thus, systems that manipulate human behavior to circumvent the will of users or systems that do not respect fundamental rights should clearly be considered as inadmissible risks. Likewise, systems incorporating technologies applied in

critical infrastructures, security components, essential public services, administration of justice, biometric identification systems, etc., seem to be at the high-risk level. However, the range is very wide, and many applications with risk tha tis in principle limited or minimal (conversational robots, emotion detection systems, spam filters, etc.) should also be monitored. The other main task is, once the regulation is determined, to define the terms of governance and the sanctioning regime.

**Author Contributions:** Formal analysis, R.F.R.F.; Investigation, L.M.G.G.; Methodology, R.F.R.F.; Writing—original draft, L.M.G.G.; Writing—review & editing, R.F.R.F. All authors have read and agreed to the published version of the manuscript.

**Funding:** This research received no external funding.

**Institutional Review Board Statement:** Not applicable.

**Informed Consent Statement:** Not applicable.

**Data Availability Statement:** Not applicable.

**Conflicts of Interest:** The authors declare no conflict of interest.

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
