# Peer review of "Digital Transformation and Artificial Intelligence Applied to Business: Legal Regulations, Economic Impact and Perspective"

_laws_

Round 1
Reviewer 1 Report
The problem analysed by the authors is, without a doubt, actual and needed. Unfortunately, the authors seem to cover all the areas of AI application and write something about every imaginable problem. This dilutes the text.
On the structural level, the author(s) try to adopt the IMRAD model, and they do it in the most surprising way imaginable. Instead of presenting an introduction with a literature review (including research questions), methods used in the study (how did we measure/analyse the data etc.), results (self-evident) and discussion (what are the conclusions and what can be researched in the future), the authors make an overview of the literature on AI and presentation of the methods used in the AI research.
Relevant legislation, including the recent European AI regulation proposal (https://eur-lex.europa.eu/legal-content/EN/TXT/?qid=1623335154975&uri=CELEX%3A52021PC0206) is missing.
We get a general overview of AI applications in business, with a hint that it could be interesting to lawyers. Basically, it can be interesting to the readers completely unfamiliar with the AI, business and law developments. The paper could be significantly improved by adding the legal component, i.e. by showing, how the law deals with the problems posed by the AI at present, is this regulation adequate and what can or should be improved and how. The authors could, for instance, do it at the end of each subchapter presenting the business application of the AI or add a separate chapter.
The IMRAD structure does not fit into the model (which is true for most black-letter legal analysis), but a short paragraph on the methods applied by the authors would be a welcome addition. I think that the literature review should focus more on what has been written on the regulation of AI in the business context. For the rest of the paper, I suggest a simple structure proposed above, i.e. presentation of a possible application in the business context, legal regulation, possible threats, possible solutions. This will make the text more accessible to readers, and will make it easy to go from "what we are going to research" through "what are the results of our research" to "what are the conclusions and proposals of legislative reforms."
Anyway, this text has potential, and I encourage authors to revise it and resubmit.
Author Response
Thank you very much for your contributions, which have undoubtedly contributed to the improvement of the work carried out. Apologies for the excessive time taken to take all the issues into consideration.
We believe that a substantial restructuring of the work has been carried out following the contributions. We will try to respond according to the issues raised:
- Introduction:
As the reviewer rightly points out, an attempt has been made in the introduction to relate one of the main objectives of the paper: linking the reality of digital transformation and artificial intelligence to regulatory needs. In this sense, the legislation referred to in Data Protection EU 2016 (lines 35-40) has been introduced as an initial basis. Similarly, the need to regulate data protection in the field of different AI models (lines 71-77) has been raised (lines 112-128). The bibliography used has been considerably expanded (lines 139-147). As pointed out by the reviewer, a final section has been included in which the objectives of the work are highlighted once the first section has been completed (lines 169-176).
- Literature Review
As indicated in the previous section, the number of bibliographical references in the work has been increased from a total of 37 (we are talking about the complete work) to 78. In this same section, as indicated by the reviewer, the recent European AI regulation proposal (https://eur-lex.europa.eu/legal-content/EN/TXT/?qid=1623335154975&uri=CELEX%3A52021PC0206) has been included (lines 236-265). In this section, the latest regulation in force has been analysed, as well as academic contributions on the subject.
- Materials and Methods
The initial approach has been restructured by including references. As the reviewer points out, ("the paper could be significantly improved by adding the legal component, i.e. by showing, how the law deals with the problems posed by the IA at present, is this regulation adequate and what can or should be improved and how. The authors could, for instance, do it at the end of each subchapter presenting the business application of the IA or add a separate chapter"), references on the regulation of IA in the business context, as well as possible threats and solutions have been taken into account. In this way, an attempt has been made to improve the structure of the paper, trying to adapt the IMRAD structure in a theoretical paper. In this sense, applications of AI to the business context have been included (lines 350-358) and two specific sections have been included (3.2. and 3.3.) referring to "Biases and Explainability" (lines 454-533) and "Ethical use of AI" (lines 534-584).
- Results and Conclusions
Despite the difficulty of presenting concrete results in a more theoretical and legal paper, an attempt has been made to raise the ethical and regulatory dilemma within business forecasts (lines 657-672). In the same way, we have tried to link the conclusions with the objectives set out in the work, marking the limitations of the same (lines 749-787).
In short, we hope that we have been able to provide the considerations indicated, and we thank you once again for the possibility of improving the work. All new contributions are listed in the text in yellow to make it easier to locate them. Thanks again
Reviewer 2 Report
I enjoyed reading the article. My comments are below. My most serious concern is that there is not enough discussion of law and AI despite that seemingly the focus of the article. There is no discussion on laws regulating AI or liability for AI acts under existing tort or contract law. There is lots of interesting literature out there. A quick good start may be Wikipedia Regulation of AI, this will lead to primary sources.
I would start with does AI exist and if so what is it. This is debatable. It seems that like the Turing Test, every time a computer can pass the test we change the definition. For example, computers will never beat humans at chess, Go, etc. I would explore this briefly if the contention is that "AI" is qualitatively different than previous technological leaps. I would think about AI creating patentable inventions, almost winning literary prizes, and fine art. May be tie this together to self-learning and 3D printing.
I would also drop a few lines about the potential of quantum computing on AI as well as the criticisms of AI and perceived limitations without quantum computing. Also, the difference types of human intelligence not all of which fit into the author's AI model.
The author spends several pages on the importance and significance of AI; however, there is little or no discussion of the role of law or regulation between the introduction and the conclusion. How have existing laws effect AI in business? Or how has the absence of law effected it.
Lines 37-38 "the legislation that governs all digital...." I am not sure what this legislation is, I would suggest that it be named or cited to or clarify that the reference here is only to EU/Spanish laws.
Lines 65-66--please develop this idea. The sentence just hangs there as if it were to transition to another idea?
Line 91 please check spelling "elabotacion" I think it should be elaboration.
Lines 97-103 Please think about adding citations. Also think about explaining the concept of AI Winter and the cycles of AI, because of over expectations.
Lines 204-245 please think about developing and more citations. I would also think about more concrete real-world examples or products.
Lines 269-73--please make this stand out a bit more it.
Lines 276-328--please define terms and develop. This section is too dense for the average reader. for example 300-301. Also, I would look more deeply into machine learning.
Lines 328-343 what "reward" to you provide an AI program as an incentive. This is very interesting, but it is not developed, and I honestly am not sure where the author(s) are going with it. I understand the concept of operant conditioning or Pavlovian conditioning in biological creatures but no AI.
Lines 503-517 go back to the interesting ideas raised in the introduction, but then are not sufficient discussed until the conclusion.
Author Response
Thank you very much for your contributions, which have undoubtedly contributed to the improvement of the work carried out. Apologies for the excessive time taken to take all the issues into consideration.
We believe that a substantial restructuring of the work has been carried out following the contributions. We will try to respond according to the issues raised. In this sense, we will try to develop the contributions made according to the indications received.
- My most serious concern is that there is not enough discussion of law and AI despite that seemingly the focus of the article. There is no discussion on laws regulating AI or liability for AI acts under existing tort or contract law. There is lots of interesting literature out there. A quick good start may be Wikipedia Regulation of AI, this will lead to primary sources.
As the reviewer rightly says, efforts have been made to correct this. Thus, an attempt has been made in the introduction to relate one of the main objectives of the paper: linking the reality of digital transformation and artificial intelligence to regulatory needs. In this sense, the legislation referred to Data Protection EU 2016 (lines 35-40) has been introduced as an initial basis. Similarly, the need to regulate data protection in the field of different AI models (lines 71-77) has been raised (lines 112-128). The bibliography used has been considerably expanded (lines 139-147). As pointed out by the reviewer, a final section has been included in which the objectives of the work are highlighted once the first section has been completed (lines 169-176). En el apartado “Literature Review”, as indicated in the previous section, the number of bibliographical references in the work has been increased from a total of 37 (we are talking about the complete work) to 78. In this same section, as indicated by the reviewer, the recent European AI regulation proposal has been included (lines 236-265). In this section, the latest regulation in force has been analysed, as well as academic contributions on the subject.
- I would start with does AI exist and if so what is it. This is debatable. It seems that like the Turing Test, every time a computer can pass the test we change the definition. For example, computers will never beat humans at chess, Go, etc. I would explore this briefly if the contention is that "AI" is qualitatively different than previous technological leaps. I would think about AI creating patentable inventions, almost winning literary prizes, and fine art. May be tie this together to self-learning and 3D printing.
As indicated by the reviewer, an attempt has been made to include the considerations made. In the Introduction section the reflection on the debate that raises the concept of intelligence vs. reasoning based on the Turing Test and aspects that differentiate AI from other technologies such as self-learning capacity (lines 112-123). In the point “Materials and Methods”, the initial approach has been restructured by including references on the regulation of IA in the business context, as well as possible threats and solutions have been taken into account. In this way, an attempt has been made to improve the structure of the paper. In this sense, applications of AI to the business context have been included (lines 350-358) and two specific sections have been included (3.2. and 3.3.) referring to "Biases and Explainability" (lines 454-533) and "Ethical use of AI" (lines 534-584).
- I would also drop a few lines about the potential of quantum computing on AI as well as the criticisms of AI and perceived limitations without quantum computing. Also, the difference types of human intelligence not all of which fit into the author's AI model.
As indicated by the reviewer, is included in the Introduction section the exponential potential of AI powered by quantum computing (lines 139-147)
- The author spends several pages on the importance and significance of AI; however, there is little or no discussion of the role of law or regulation between the introduction and the conclusion. How have existing laws effect AI in business? Or how has the absence of law effected it.
Despite the difficulty of presenting concrete results in a more theoretical and legal paper, an attempt has been made to raise the ethical and regulatory dilemma within business forecasts (lines 657-672). In the same way, we have tried to link the conclusions with the objectives set out in the work, marking the limitations of the same (lines 749-787).
- Lines 37-38 "the legislation that governs all digital...." I am not sure what this legislation is, I would suggest that it be named or cited to or clarify that the reference here is only to EU/Spanish laws.
The legislation to which it refers is specified at the European level
- Lines 65-66--please develop this idea. The sentence just hangs there as if it were to transition to another idea?
As the reviewer indicates, the idea has been developed and linked to data-driven Artificial Intelligence projects (lines 71-77).
- Line 91 please check spelling "elabotacion" I think it should be elaboration.
Thank you very much, the error has been corrected
- Lines 97-103 Please think about adding citations. Also think about explaining the concept of AI Winter and the cycles of AI, because of over expectations.
Attempts have been made to explain the reviewer's point (lines 124-128).
- Lines 204-245 please think about developing and more citations. I would also think about more concrete real-world examples or products.
The contrast of these use cases with real references has been included including many more citations (lines 236-265 and epigraph 3)
- Lines 269-73--please make this stand out a bit more it.
An attempt has been made to elaborate on the explanation and meaning of the four points mentioned (lines 350-358)
- Lines 276-328--please define terms and develop. This section is too dense for the average reader. for example 300-301. Also, I would look more deeply into machine learning.
Further references have been included to develop more information on the algorithms.
- Lines 328-343 what "reward" to you provide an AI program as an incentive. This is very interesting, but it is not developed, and I honestly am not sure where the author(s) are going with it. I understand the concept of operant conditioning or Pavlovian conditioning in biological creatures but no AI.
Included reference for more information about reinforced learning in AI
- Lines 503-517 go back to the interesting ideas raised in the introduction, but then are not sufficient discussed until the conclusion.
Added “3.2. Biases and explainability” and “3.3. Ethical use of Artificial Intelligence” (lines 454-584) and new results and conclusions (lines 657-672, 749-787).
In short, we hope that we have been able to provide the considerations indicated, and we thank you once again for the possibility of improving the work. All new contributions are listed in the text in yellow to make it easier to locate them. Thanks again
Reviewer 3 Report
This manuscript is intended for the special issue of Laws on Legal-Economic Issues of Digital & Collaborative Economy and intends to show according to the authors own word that ‘‘digital transformation is no longer a buzzword; it is a fact.’’ From a purely grammatical standpoint, the article is generally well written, however it lacks the proper structure that one would expect in a scholarly manuscript. For example, the introduction does not clearly provide the objective of the paper to the reader (I found them in the conclusion), the section entitled ‘literature review’ does not read like a review and relies on less than 10 sources; the result section does not provide any academic research results… etc. Other problems of the article, includes the poor quality of figures provided (this can probably be easily fixed) and the completely uncritical perspective of AI provided throughout the manuscript.
The position of the author(s) is that AI is extremely beneficial to the economy and therefore that it should not be slowed down by the regulatory framework. This would be fine if the authors had acknowledged the limit of this perspective (does not consider the considerable ethical and social issues raised by AI) and acknowledged the important body of work of scholars providing a different opinion. However, the article does not do any of this instead choosing to stay at a very superficial descriptive level, explaining the administrative changes facilitated by AI and the economic benefit they entail. The resulting manuscript reads like an intro to AI and the marketplace for beginner. Given this it seems more fitted for an industry brief than for a scholarly legal journal. Lastly, the manuscript addresses the topic of legal policies (in the context of AI) only in an extremely general manner which can be summed as: law must not hinder the integration of AI in the workplace, instead it should be used to foster international harmonisation of AI practices. Again, this sort of general affirmation would be fine for a very short information blog on the topic, but it is insufficient for a scholarly publication focussed on law.
Author Response
Thank you very much for your contributions, which have undoubtedly contributed to the improvement of the work carried out. Apologies for the excessive time taken to take all the issues into consideration.
We believe that the work has been substantially restructured in line with the contributions you have made. We will try to respond according to the issues that were raised.
- Introduction:
As the reviewer rightly points out, an attempt has been made in the introduction to relate one of the main objectives of the paper: linking the reality of digital transformation and artificial intelligence to regulatory needs. In this sense, the legislation referred to in Data Protection EU 2016 (lines 35-40) has been introduced as an initial basis. Similarly, the need to regulate data protection in the field of different AI models (lines 71-77) has been raised (lines 112-128). The bibliography used has been considerably expanded (lines 139-147). As pointed out by the reviewer, a final section has been included in which the objectives of the work are highlighted once the first section has been completed (lines 169-176).
- Literature Review
As indicated in the previous section, the number of bibliographical references in the work has been increased from a total of 37 (we are talking about the complete work) to 78. In this same section, we have tried to analyse the current regulation on AI (lines 236-265). In this section, we have analysed the regulations as well as academic contributions in this respect.
- Materials and Methods
The initial approach has been restructured by including references. References on the regulation of AI in the business context, as well as possible threats and solutions, have been taken into account. In this way, an attempt has been made to improve the structure of the paper, trying to adapt the IMRAD structure in a theoretical paper. In this sense, applications of AI to the business context have been included (lines 350-358) and two specific sections have been included (3.2. and 3.3.) referring to "Biases and Explainability" (lines 454-533) and "Ethical use of AI" (lines 534-584). In these sections we have tried to question the ethical and regulatory considerations of the concept of AI applied to business, assessing it from a normative point of view.
- Results and Conclusions
Despite the difficulty of presenting concrete results in a more theoretical and legal paper, an attempt has been made to raise the ethical and regulatory dilemma within business forecasts (lines 657-672). In the same way, we have tried to link the conclusions with the objectives set out in the work, highlighting its limitations and the regulatory implications that AI systems must take into account (lines 749-787).
In short, we hope that we have been able to provide the considerations outlined above, and we thank you once again for the possibility of improving the work. All new contributions have been highlighted in yellow to make them easier to locate. Thank you again.
Round 2
Reviewer 1 Report
The authors have made an attempt to improve the paper, unfortunately this attempt is not good enough. Principal problem - the legal component is still insufficient and the authors seem to think that they are the first ones to deal with AI & Law. It is not true. There is a lot of legal research going on which they neglect to mention. The "Artificial Intelligence and Law" journal has some 20 years history. The authors quote some primary legal documents (good), however a) they quote "as is" - without reference to previous research done on the subject and b) they quote EU materials somewhat surprisingly in Spanish version, although some 27 others, including English are available. The golden rule is that you should make references to documents in the language of the paper if they are available. Multiple versions of the same document are sometimes quoted in the legal literature, e.g. when Spanish and English versions differ significantly. This does not seem to be the case. Price Waterhouse is a fine company, but their Fiscal and Legal Periscope can hardly be called the seminal source on AI law.
The analytical part is still missing - the authors do not state clearly, what has already been don in the area, what is their contribution to science and to what extent their work adds something to the general sum of knowledge.
Author Response
First of all, many thanks again for the contributions and the time taken to improve the work. We have tried to include all the indications pointed out by the reviewer:
- A special effort has been made to improve the legal references of the work, Thus, both in the section "Literature Review" and in "Results" academic references from different authors on the legal and ethical aspects of the subject to be treated have been included (lines 218-223, 242-271, 731-739). More than 22 new references have been included in these extensions.
- As indicated by the reviewer, many of the new materials referenced come directly from the EU in their initial description, i.e., not in their Spanish version.
- Finally, we have tried to provide in the "Results" section the contributions and conclusions expected from the present work (lines 629-645, 731-747).
We hope to have taken into account the indications made in this second revision.
Reviewer 3 Report
This is my second review of this article on AI, business and regulation. The authors did put in a good effort to respond to the comments of my first review. This generally result in a better documented manuscript with a tighter structure (still a bit too convoluted, but acceptable). One remaining problem is that the new additions were made by someone who is not a native English writer. Thus sentence structures often do not meet language requirements and grammatical mistakes abound. I will give a few examples of those below, but I did not attempt to comprehensively note all of them. If it is accepted, all changes will need to be thoroughly reviewed to make sure the writing style is suitable for an academic publication. Finally, the objective of the ‘results’ section shouldn’t be to ‘‘highlight the importance of the concepts of digital transformation and artificial intelligence linked to business’’ (as stated by authors) but to present the result of their literature review. If this is not the case, perhaps the title of the section should be changed from ‘results’ to something more appropriate.
Example of English that needs to be corrected in the review:
- Use of ‘the’ in front of plural nouns
- Capital letter used at beginning of common words
- Line 39 ‘varios’ should spell ‘various’
- Line 171 to 173 = runabout sentence, double use of word ‘regulation’ is illogical, please
review sentence structure
- Line 456, please clarify meaning of ‘has now become a law’. Is the author speaking literally (if so what law precisely?) or did he meant to say ‘a legal issue’?
- Line 462-463 and 475-77 appear contradictory, please clarify
- Line 455-586 comport multiple issues of odd or non-conform sentence structure in the English language making the reading of the section particularly challenging for readers.
- 751 ‘hig-risk escenario’ should be ‘high risk scenario’
Author Response
First of all, many thanks again for the contributions and the time taken to improve the work. We have tried to include all the indications pointed out by the reviewer:
- A special effort has been made to improve the legal references of the work. In the section "Literature Review" and in "Results" academic references from different authors on the legal and ethical aspects of the subject to be treated have been included (lines 218-223, 242-271, 731-739). More than 22 new references have been included in these extensions. As indicated by the reviewer, was intended to emphasize the importance not only of highlighting the concepts of digital transformation and artificial intelligence linked to the enterprise, but also to present the results of the bibliographic review carried out.
- Finally, we have tried to provide in the "Results" section the contributions and conclusions expected from the present work (lines 629-645, 731-747).
- Finally, the grammatical indications pointed out by the reviewer have been corrected.
We hope to have taken into account the indications made in this second revision.
Round 3
Reviewer 1 Report
Dear Authors,
It is much better than in the beginning. Still, the text is descriptive rather than analytical, but having taken into account the topic - readers can benefit from it, especially if they are new to the subject.